# Effect of Egg Washing and Hen Age on Cuticle Quality and Bacterial Adherence in Table Eggs

**DOI:** 10.3390/microorganisms12102027

**Published:** 2024-10-08

**Authors:** Garima Kulshreshtha, Cian Ward, Nicholas D. Calvert, Cristina Benavides-Reyes, Alejandro B. Rodriguez-Navarro, Ty Diep, Maxwell T. Hincke

**Affiliations:** 1Department of Cellular and Molecular Medicine, Faculty of Medicine, University of Ottawa, Ottawa, ON K1H 8M5, Canada; gr784654@dal.ca (G.K.); cward72@uwo.ca (C.W.); ncalvert101@gmail.com (N.D.C.); 2Departamento de Mineralogia y Petrologia, Universidad de Granada, Campus de Fuentenueva, 18002 Granada, Spain; crisbr@ugr.es (C.B.-R.); anava@ugr.es (A.B.R.-N.); 3Lyn Egg Production and Grading, Burnbrae Farms Limited, 75 Laurier Ave E, Ottawa, ON K1N 6N5 Canada; tdiep@burnbraefarms.com; 4Department of Innovation in Medical Education, Faculty of Medicine, University of Ottawa, Ottawa, ON K1H 8M5, Canada

**Keywords:** *Salmonella Typhimurium*, eggshell cuticle, hen age, commercial egg washing, bacterial adherence, confocal microscopy

## Abstract

The cuticle covering the outer surface of an eggshell functions as both a physical and chemical barrier against invading microorganisms. Contamination of eggs by microbial pathogens progresses in four stages: bacterial attachment to the egg surface, penetration through the cuticle and eggshell, multiplication within the underlying membranes, and the final stage of contaminating the egg contents. Therefore, it is important to study bacterial count at the first point of contact, i.e., on the surface of the eggs. In this study, we have evaluated the impact of differences in cuticle quality (due to egg washing and hen age) on bacterial load. We compared bacterial adherence on the eggshell surface of white eggs which were either washed (graded) or unwashed (ungraded), collected from Lohmann laying hens of different ages: early (24–28 weeks), mid-lay (44–48 weeks), and late (66–70 weeks). We aimed to determine the impact of hen age and egg washing on differences in cuticle quality and bacterial adherence. Our results indicate that hen age (up to 70 weeks) and commercial egg washing do not significantly impact bacterial adherence on eggshell surfaces. We have developed a novel method using green fluorescent protein (GFP)-expressing *Salmonella typhimurium* to estimate adherence of bacteria to the eggshell surface, with independent measurement of autofluorescence to quantitate cuticle deposition. *S. typhimurium* were localized, adhering to cracks visible on the outer cuticle in ungraded eggs, indicating that egg-associated pathogens usually enter the egg interior either through respiratory pores in eggshells or through shell micro-cracks. The results of this study can be utilized to optimize innovative methods for predictive microbiology in order to achieve egg safety.

## 1. Introduction

Foodborne microbial pathogens lead to recalls and outbreaks, resulting in economic losses, illnesses, hospitalization, and deaths. The CDC estimates that *Salmonella* is a leading microbial contaminant in eggs and egg products, which has been associated with product withdrawal, recalls, and health concerns [1,2,3]. In North America, eggs are washed and graded before sale; washing ensures the cleaning and removal of potential pathogens from the eggshell surface in order to maximize the quality and safety of table eggs [4,5]. The grading process involves washing the egg surface and evaluating for factors including shell quality, cleanliness, weight, and interior quality [4,5]. In other jurisdictions (Europe and Asia), eggs are not washed before retail sale, and the exterior status of the egg is critically important to prevent contamination [6]. Nonetheless, some persistent pathogens, including *Salmonella* Enteritidis and *E. coli,* continue to be introduced into food supply chains via contaminated eggs [7]. For example, in 2020–2021, an outbreak of *Salmonella* caused by consumption of contaminated poultry in the Atlantic provinces of Canada resulted in 70 cases of salmonellosis and 19 hospitalizations [8]. According to the Centers for Disease Control and Prevention (CDC), *Salmonella* causes the hospitalization of about 27,000 individuals and an estimated 420 deaths each year in the United States [9]. Globally, foodborne *Salmonella* bacteria cause over 80.3 million infections per year, with contaminated poultry being a major source of these illnesses [10].

The cuticle covering the outer surface of the eggshell functions as both a physical and chemical barrier against invading microorganisms. The antimicrobial activity of the cuticle is due to its protein components such as Ovocalyxin32, Ovocalyxin-36, Cystatin, Ovoinhibitors, Ovotransferrin, and Ovocleidin-17 [11,12]. There are two primary mechanisms for contamination of an egg by *Salmonella*: 1. vertical transmission, which originates from infected ovaries of a laying hen via the transovarial route, and the egg is contaminated before being laid [13]; 2. horizontal transmission, in which a pathogen transpires through the eggshell via the pores or physical defects such as cracks/micro-cracks, resulting in egg contamination [14,15]. Several other serovars of *Salmonella,* such as Typhimurium, Infantis, and Newport, have been isolated from eggshell surfaces [16]. Contamination of eggs by microbial pathogens progresses in four stages: bacterial attachment to the egg surface, penetration through the cuticle and eggshell, multiplication within the underlying membranes, and the final stage of contaminating the egg contents [17]. Therefore, it is important to study bacterial count at the first point of contact, i.e., on the surface of the eggs, not just the egg’s internal contents. There is a lack of research regarding the level of eggshell surface contamination by major bacterial pathogens such as *Salmonella*.

Poor cuticle coverage increases the susceptibility of eggs to contamination by bacterial pathogens, including *Salmonella*. The age of the hen is an important factor in establishing egg safety. It has been demonstrated that table eggs from ageing older hens display a poorer eggshell quality and are more prone to contamination by bacteria [18,19]. Previously, it has been shown that hen age (up to 49 weeks) significantly affected total bacterial accumulation on the eggshell surface and in shell pores of eggs from free-range housing environments [17]. Another study reported an increase in the amount of lysozyme and ovotransferrin (antimicrobial proteins in egg white) with hen age (up to 100 weeks) [20]. Studies reporting poultry trials from flocks of different bird strains and ages and rearing conditions increase the probability of finding conflicting evidence due to enhanced variability. Other factors, including housing, flock size, flock, stocking density, farm management practices and programs, infection due to pests, stress, and hygiene, are critical in determining the level/type of bacteria present in the eggshell, pore, and internal contents [17].

We have demonstrated that the hen age, strain, and commercial washing process of eggs influence the cuticle composition and quality. The industrial washing process can remove some of the cuticle from the outer eggshell surface; nevertheless, the cuticle plug proteins which cover the pore openings remain intact in order to block bacterial entry via the respiratory pores. Elemental analysis showed that the pore inner surface is rich in phosphorus and chemically distinct from the bulk eggshell mineral. FTIR analysis revealed that cuticle chemical composition is affected by hen age, strain, housing system, and egg washing [21,22].

In this study, we have further studied the impact of differences in cuticle quality (due to egg washing and hen age) on bacterial load. We compared bacterial adherence on the eggshell surface of white eggs which were either washed (graded) or unwashed (ungraded), collected from Lohmann laying hens of different ages: early (24–28 weeks), mid-lay (44–48 weeks), and late (66–70 weeks). Additionally, we have also evaluated the influence of these factors on the surface hydrophobicity of table eggs. Our study concluded that hen age (up to 70 weeks) and commercial egg washing do not significantly impact bacterial adherence on eggshell surfaces.

## 2. Materials and Methods

### 2.1. Source of Eggs

Eggs were provided by Burnbrae Farms Limited (Lyn, ON, Canada) from Lohmann hens at the following ages: early (24–28 weeks), mid (44–48 weeks), and late (66–70 weeks). The eggs were either unwashed/ungraded or subjected to commercial egg washing as described previously [21,22] and designated as washed or graded eggs (throughout the manuscript). 

### 2.2. Chemicals and Instruments

Thirty eggs were selected from each group for analysis of surface hydrophobicity. Of these, 18 eggs from similar age groups were used for the measurement of bacterial adherence by a microbiology technique. Additionally, 18 eggs were randomly selected for measurement of bacterial adherence by SEM, and 12 eggs were collected for estimation of cuticle and bacterial adherence by confocal microscopy. The eggs were visually examined, and only intact eggs with no pinholes or cracks were used in the experiments. The internal contents of each egg were removed using a standardized method described previously [21,22]. Only the interior of the egg was rinsed in order to avoid washing the outer egg surface. Eggs were stored at −20 °C before experimental analysis and were completely randomized before analysis in order to avoid bias. 

The chemicals were purchased from Sigma-Aldrich (Oakville, ON, Canada) or ThermoFisher Scientific (Waltham, MA, USA). The cuticle from the outer eggshell surface was removed by bleach treatment using a standardized protocol as described previously [21]. Bleach-treated eggshells were used as a control in all the experiments. Removal of cuticle after bleach treatment was validated by MST cuticle blue staining using MST cuticle blue (MS Technologies Ltd., Northamptonshire, UK). 

### 2.3. Surface Hydrophobicity 

The surface hydrophobicity (contact angle) on the surface of the eggshells was measured using the sessile drop technique as described previously [21]. A total of 24 eggs were analyzed for each experimental unit/condition.

### 2.4. Cell Attachment Assay to Evaluate Bacterial Adherence

#### 2.4.1. Salmonella and Bacillus Strains and Culture Conditions

The green fluorescent protein (GFP)-expressing and antibiotic-resistant Gram-negative *Salmonella typhimurium* str. SL1344 was provided by Drs. S. Sad and R. Russell (Faculty of Medicine, University of Ottawa). A Gram-positive strain of *Bacillus cereus* (ATCC 11778) was obtained from the University of Ottawa (Ottawa, ON, Canada). Luria–Bertani (LB) broth or agar (BioShop, Canada) with 100 μg/mL ampicillin was utilized for the growth and maintenance of *Salmonella*, while LB agar or broth without antibiotics was used for *Bacillus* cultures. Both bacterial strains were revived and grown from glycerol stocks by streaking on LB agar, as described previously [21]. The bleach-treated eggshells with no cuticle were used as a control in all bacterial cell attachment assays analyzed using both microscopic and microbiology-based methods.

#### 2.4.2. Estimation of Cuticle and Salmonella Adherence by Confocal Microscopy

*Salmonella* adherence and cuticle coverage were determined on cross-fractured eggshell fragments from the sharp, blunt, and equatorial regions (4 fragments/region) of every egg (*n* = 4 frames/fragment; size = 10–15 mm^2^) using confocal microscopy. This method was developed and used for the first time in this study.

##### Sample Preparation

The bacterial suspension in PBS (1 mL, optical density OD = 0.2 at 600 nm) was incubated with eggshell fragments (*n* = 4 × 3 = 12 eggs/experimental unit or condition) in a 12-well microplate for 3 h at 37 °C in a rotating incubator at 150 rpm. Next, the eggshell fragments were washed with PBS to get rid of non-adhering cells of *Salmonella*. All samples were then fixed with 4% paraformaldehyde (PFA) for 10 min at 4 °C. The PFA was then removed from each well, and eggshell fragments were rinsed 3 times with PBS to remove any excess PFA. The ES samples were left in a laminar hood until dry. Once dried, ES samples were kept at 4 °C until further processing. Eggshell samples, either with or without 4% PFA-fixed bacteria, were mounted on glass slides using coverslips (Thermo Fisher Scientific, Waltham, MA, USA) with a 1 mm rubber spacer attached to them.

##### Image Acquisition

Images of ES samples were acquired using a Zeiss LSM 880 with an AiryScan confocal microscope (Zeiss, Oberkochen, Germany) through a 20× objective. Figure 1 provides an overview of the acquisition process and the path of the excitation and emission light. Samples with or without bacteria were imaged using profiles of excitation and emission spectra for green fluorescent protein (eGFP) and mCherry (green and red, respectively). Two channels were used to visualize the ES surface: a) a red channel used to image the autofluorescence of the ES surface, and b) a green channel used to detect the GFP bacteria adhered to the ES surface (Figure 2). The red channel focused a laser with a wavelength of 561 nm on an ES fragment, and the ES autofluorescence that was in the range of 582–625 nm was captured by the microscope. The green channel used a laser with a wavelength of 488 nm as excitation light, while the emission spectrum measured by the CLSM was in the range of 493–567 nm. Both tile-scanning and Z-stacking were used for image acquisition. Multiple images of ES were captured at different vertical distances from the objective at more than one focal plane and grouped as a Z-stack in order to visualize the entire curved surface of the eggshells. One single image at a specific z-interval is called a z-slice. To increase the area imaged, four adjacent fields of view per z-slice known as ‘titles’ were stitched together using Zen Blue Software (Version 2.3). Each tile has dimensions of 425.1 μm × 425.1 μm, leading to a total area of the composite image of 0.7228 mm^2^. All representative images and videos were rendered for display into 3-dimensional models using Imaris software (Version 9.8, Bitplane, Zurich, Switzerland). 

##### Image Processing

(a)Estimation of cuticle: ES confocal images were processed to determine the relative fluorescence in both the green and red channels after image acquisition through ImageJ (FIJI, NIH) processing. Raw .czi files acquired from confocal imaging had a Z-projection of the summed slices created for each channel. The integrated density for each channel was then measured. This process was performed through an in-house script so that large dataset processing could be automated (Appendix A).(b)Estimation of bacterial adherence: ES images with bacterial adherence were processed with the green channel after image acquisition through ImageJ (FIJI, NIH) processing. Raw .czi files acquired from confocal imaging had a Z-projection of the summed slices created for the green channel, followed by a rolling ball subtraction to remove any background fluorescence (size 5 µm). A threshold using the Yen method was applied to the image to create a mask. This mask was applied back to the Z-projection before the threshold, such that only the green spots associated with the GFP-expressing bacteria were visible on the image. Integrated density (IntDen) was then measured, with the applied mask ensuring that the only measured signal came from the GFP-expressing bacterial spots. This process was performed through an in-house script so that large dataset processing could be automated (Appendix A). IntDen is the product of the mean pixel value and the area of the selected region. The relative quantity of bacteria and cuticle on the eggshell surface was quantified and represented as the IntDen.

#### 2.4.3. Determination of *Bacillus cereus* Adherence by Scanning Electron Microscopy (SEM)

A cell attachment assay was performed to determine whether the cuticle on the eggshell surface prevents adherence of *Bacillus cereus* cells using a method described previously [23,24] with some modifications. Briefly, small squares (1 cm^2^) of eggshells were immersed in bacterial suspension in PBS (1 mL, optical density OD = 0.2; ∼10^8^ CFU mL^−1^ at 600 nm) for 3 h at 37 °C. Next, the eggshell fragments were washed with PBS to get rid of non-adhering cells of *Salmonella*. All samples were then fixed with 4% PFA for 10 min at 4 °C, and the number of adhering bacterial cells was counted (SEM; TeScan Vega-II XMU, Brno—Kohoutovice, Czech Republic). Bacterial cell counts were performed on five random locations across each shell piece, and results were represented as bacterial cell density per μm^2^ of the shell surface.

#### 2.4.4. Determination of Salmonella Adherence by a Microbiology Technique

A previously established cell attachment assay based on a microbiology technique was used to evaluate adherence of *Salmonella* on the outer surface of eggshells [21]. 

### 2.5. Statistical Analysis

Statistical analyses used R Statistical Software (Version 3.2.4; R Development Core Team, Vienna, Austria) and MINITAB 17 (Minitab, LLC, State College, PA, USA) with the level of significance of *p* < 0.05 for all experimental work, as described previously [21,22]. Confocal measurements were analyzed using a Kruskal–Wallis test and Dunn’s post hoc test. 

## 3. Results

### 3.1. Estimation of Cuticle Using Fluorescent Imaging

Cuticle assessment using fluorescent imaging confirmed that the bleach treatment effectively removed (*p* < 0.05, *n* = 12) the cuticle present on the outer surface of eggshells (Figure 3A,D). Areas of high red or yellow fluorescence indicate the presence of cuticle. Lower integrated density (IndDen) was detected in the shells from eggs treated with bleach (0.48 ± 0.86%, *n* = 12) in comparison to the eggshells of the graded or ungraded eggs (11.36 ± 7.01% and 12.86 ± 2.48%, *n* = 12), supporting the conclusion that the bleach treatment is effective in the removal of the outer-surface cuticle.

Although we did not observe any significant differences in the IndDen of the surface cuticle in graded vs. ungraded eggs, a trend of higher cuticle was observed in the ungraded as compared to the graded eggs (Figure 3B–D).

### 3.2. Hydrophobicity of Eggshell Surface

In our previous study, we demonstrated the importance of the cuticle proteins to maintain hydrophobicity on the outer surface of the eggshell [21]. In the present study, we observed that the contact angle of the surface of the ungraded eggs (97.11 ± 9.03°) was significantly higher (*p* < 0.05, *n* = 72) than graded (92.51 ± 8.67°) and bleach-treated eggs (43.02 ± 9.16°) (Figure 4B). Even though we observed a trend of higher contact angles in the ungraded eggs from hens aged 24–28 weeks (95.62 ± 10.12°), 44–48 weeks (96.90 ± 8.35°), and 66–70 weeks (98.82 ± 8.64°) compared to graded eggs from hens aged 24–28 weeks (92.36 ± 8.95°), 44–48 weeks (93.63 ± 8.20°), and 66–70 weeks (91.55 ± 9.08°), this was not statistically different (*p* > 0.05) (Figure 4A).

### 3.3. Adherence of Bacillus cereus on the Surface of Eggshells

The bleach-treated eggshells without cuticle showed significantly higher (*p* < 0.05, *n* = 18) counts of adhering *Bacillus cereus* cells on the outer eggshell surface, in comparison to an intact cuticle on eggshells from graded and ungraded eggs (Figure 5). A trend of higher adherence of *B. cereus* cells was identified in graded eggs in comparison to ungraded eggs (Figure 5).

### 3.4. Adherence of Salmonella typhimurium on the Surface of Eggshells

#### 3.4.1. Quantification by Confocal Imaging

We did not observe any significant difference in adherence of *S. typhimurium* cells on the surface of eggshells from graded, ungraded, or bleach-treated eggs (Figure 6; *p* > 0.05, *n* = 12). However, a trend of more bacterial cells being deposited on the surface of bleach-treated eggs in comparison to graded and ungraded eggs was observed (Figure 6). *S. typhimurium* cells were localized close to cracks present on the outer-surface cuticle of ungraded eggshells.

#### 3.4.2. Quantification by Microbiology Technique

The bleached eggs without cuticle showed significantly higher (*p* < 0.05, *n* = 18) *Salmonella typhimurium* counts adhering to the outer surface of the eggshell in comparison to an intact cuticle on the eggshell surface of graded and ungraded eggs from hens at various ages (24–28 weeks, 44–48 weeks, and 66–70 weeks) (Figure 7A).

The graded eggs showed higher counts (*p* < 0.05) of adhering *Salmonella* cells in comparison to the ungraded eggs (Figure 7C). A trend of higher adherence of *Salmonella* cells was observed on eggshell surfaces in graded eggs in comparison to ungraded eggs from hens at all three ages (Figure 7B).

## 4. Discussion

An intact cuticle is the first line of defence against invading bacterial pathogens. The cuticle functions as an efficient physical and chemical barrier against microbial aggression in order to maintain egg safety [25,26,27]. The role of cuticle in reducing/preventing bacterial colonization and penetration has been well documented [27,28]. In our previous study, we characterized eggshell cuticle and pore plugs in eggshells in response to changes in hen age, egg colour, and commercial egg-washing process [22]. A follow-up study confirmed that the removal of the cuticle increases adhering *Salmonella typhimurium* cells on the outer surface of the eggshell, indicating the role of the cuticle in modulating bacterial adherence [21]. Here, we demonstrate the effect of egg washing and hen age on surface hydrophobicity and bacterial load on eggshell surfaces in table eggs.

In this study, we have developed a novel enumeration method using green fluorescent protein (GFP)-expressing *Salmonella typhimurium* to estimate bacterial adhesion to the eggshell surface and quantify cuticle deposition (using measurement of autofluorescence). Our results showed that the cuticle constituents reduced bacterial adherence to the outer surface of eggshells in ungraded eggs, independent of bacterial penetration. The antibacterial activity is suggested to be due to antibacterial cuticle proteins such as lysozyme C, ovotransferrin, ovocalyxin-32 (OCX-32), and ovocleidin-17 (OC-17), which constitute the basis for its antimicrobial activity [29,30,31,32]. A trend of higher bacterial load was observed on the outer eggshell surface of graded eggs compared to ungraded eggs. Next, quantification of *Bacillus cereus* adherence by cell attachment assay using scanning electron microscopy showed a similar trend of higher bacteria on graded vs. ungraded eggshell surfaces (Figure 5). Estimation of bacterial adherence by SEM is labour-intensive, expensive, and time-consuming for both sample preparation and image analysis. A culture-based system typically requires five to seven days for bacterial confirmation and enumeration. Hence, the enumeration of bacterial fluorescence on eggshell surfaces could be a consistent alternative method to determine bacterial adherence in a shorter time. 

Quantification of cuticle autofluorescence showed higher cuticle deposition on ungraded eggs compared to bleach-treated eggs, while no differences were observed between graded or ungraded eggshell surfaces (Figure 3). A previous study has shown that the colour pigment protoporphyrin IX (PPIX) remains embedded in the protein phase of brown and white eggshells as highly fluorescent monomers [33]. Moreover, there is evidence of quantification of tryptophan fluorescence from both the cuticle and the sub-cuticle matrix as a measure of cuticle deposition [34]. Further investigation is required to identify specific proteins contributing to cuticle fluorescence on the outer surface of eggshells. We also observed a higher level of variability in the percentage of cuticle proteins quantified from the outer surface of graded eggs. The previous literature has shown that cuticle deposition on the outer surface of eggshells is not uniform and can be variable between eggs. Moreover, hen age, commercial egg washing, hen strain, and housing systems can also affect cuticle deposition [21,22,32,35]. The deposition of cuticle proteins on the outer surface could be variable from one egg to another, which could contribute to higher variability in graded eggs.

The outer surface of unwashed eggs was more hydrophobic compared to washed eggs (Figure 4B). Also, both washed and unwashed surfaces with cuticle had a significantly higher contact angle than bleach-treated eggs without cuticle (Figure 4A). It is well documented that surface material layer characteristics such as wettability, texture, roughness, and surface topography play a critical role in initiating the adhesion of bacteria to surfaces [36,37]. Our previously published study established an inverse correlation between eggshell surface hydrophobicity and *Salmonella* adherence, indicating that surface cuticle proteins play an important role in reducing bacterial contamination [21]. Similarly, a previous study has shown an increased adhesion of Gram-negative bacteria to the mineral with decreased hydrophobicity [37]. Moisture/humidity present on the surface of the eggshell increases the risk of egg contamination; hence, surface hydrophobicity can be an important indicator of cuticle quality in order to determine egg safety. Measurement of contact angle could be implemented at the breeder company level to select hens for thicker cuticle. The advantages of this new method to evaluate cuticle quality are that it is 1. cost-effective, 2. rapid, 3. non-invasive, and 4. non-destructive. Such application will be helpful to breeders in the selection of eggs with a thicker cuticle.

*Salmonella* adherence was also determined using an established bacterial cell attachment assay. Although we did not observe large differences in bacterial adherence between eggs from different-aged hens (Figure 7A,B), the statistical assessment indicated that significantly (*p* < 0.05) lower bacterial counts were observed on the surface of unwashed eggs in comparison to washed eggs (Figure 7C). This evidence suggests that, in addition to blocking microbial passage through the respiratory pores, the cuticle also plays a key role in reducing bacterial adherence [34,35,38]. Moreover, the microscopic studies showed that bacterial cells were localized close to micro-cracks on the outer cuticle surface of ungraded eggshells (Figure 8). This indicates that egg-associated pathogens usually penetrate the egg interior either through respiratory pores or through the micro-cracks in eggshells. After cuticle removal by bleach treatment, higher localization of bacterial load was observed near the opening/mouth of the cuticle pore plug on the outer eggshell surface. This indicates that the cuticle deposited over pore plugs is abundant in antiadhesive proteins, which effectively reduces the attachment of bacteria to the outer shell surface in order to restrict bacterial penetration into the egg interior. Antiadhesive/antimicrobial proteins may be sufficiently located/concentrated to plug the pore rather than being uniformly distributed as a complete cuticle coverage in order to prevent pathogen entry into the egg’s contents. This hypothesis should be experimentally evaluated and needs further investigation.

We observed a higher load of Gram-negative *Salmonella typhimurium* on the outer bleach-treated eggshells compared to Gram-positive *Bacillus cereus* (Figure 5, Figure 6 and Figure 7). This could be due to differences in bacterial surface charge density between *Salmonella typhimurium* and *Bacillus cereus*. The surface charge density has been shown to affect initial bacterial attachment and subsequent biofilm formation on the material surface [36,39]. Adsorption saturation densities of Gram-negative vs. Gram-positive bacteria on the exterior surfaces are characteristically distinct. The negative charge density of the lipopolysaccharide (LPS) coated on the outer surface of Gram-negative bacteria is 7X greater than that of the protein surface layer of Gram-positive bacteria. It has been documented that the smooth surface protein S-layer of the Gram-positive cells exhibits a small adsorption density, while the rougher LPS surface of the Gram-negative cells has a significantly larger adsorption density [39]. Hence, *Salmonella* could exhibit higher affinity to the calcitic surface of eggshell in bleach-treated eggs due to the higher surface charge density. 

We did not observe the large effects of increasing hen age or commercial egg washing on bacterial adherence or surface hydrophobicity on the outer surface of eggshells (Figure 5, Figure 6 and Figure 7). However, adhering bacterial cells on the outer cuticle surface were significantly lower in all conditions when compared to bleached, treated eggs (with no cuticle) (Figure 5, Figure 6 and Figure 7). Also, in our previous study, we observed that cuticle quality declined with increasing hen age or due to commercial washing. This indicates that some cuticle components can be thinner due to increasing hen age or wear away during the commercial washing process, but the constituent antimicrobial proteins could still be sufficiently abundant and functional/active to block pathogens from adhering to the outer surface of the eggshell irrespective of hen age or egg washing, thus protecting the egg from bacterial contamination. The antimicrobial cuticle proteins are possibly more stable to the above traits and can compensate for cuticle loss in order to provide better protection to the egg against invading pathogens.

## 5. Conclusions

To conclude, this study describes novel strategies to enumerate and localize bacteria associated with eggshell surfaces. The results from this study can be utilized to optimize innovative methods for predictive microbiology in order to achieve egg safety. This study validated that the measurement of the contact angle is correlated with cuticle quality. Eggs can be categorized based on the hydrophobicity of the eggshell surface to evaluate cuticle quality in commercial egg grading systems. Hen age and commercial egg washing do not significantly impact bacterial adherence. In the United States and Canada, table eggs are commercially washed before sale; hence, our findings should maintain the confidence of consumers in eggs produced in Canada.

Currently, the laying hen breeder industry is developing the concept of the “long life” layer, able to produce up to 500 eggs during a 100-week laying cycle. Bird health and egg quality are significant criteria to achieve this target. Hence, our findings of lower bacterial load on the outer surface of eggs produced from older hens (66–70 weeks) are reassuring for quality in eggs from older hens. Further studies are required to understand correlations/links between cuticle coverage/deposition/properties and its protective/antimicrobial functions.

## Figures and Tables

**Figure 1 microorganisms-12-02027-f001:**
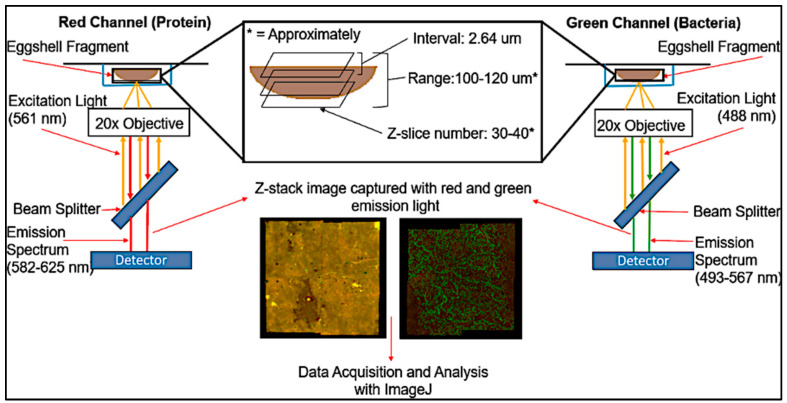
The light path of confocal imaging for estimation of cuticle and bacterial adherence.

**Figure 2 microorganisms-12-02027-f002:**
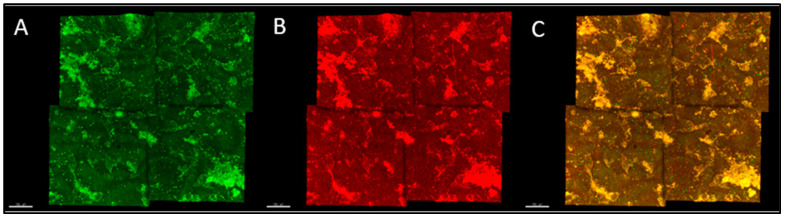
CLSM images: Green channel of the sharp end of a white ungraded ES fragment (**A**). The red channel of the sharp end of a white ungraded ES fragment (**B**). Composite image of the sharp end of a white ungraded ES fragment (**C**).

**Figure 3 microorganisms-12-02027-f003:**
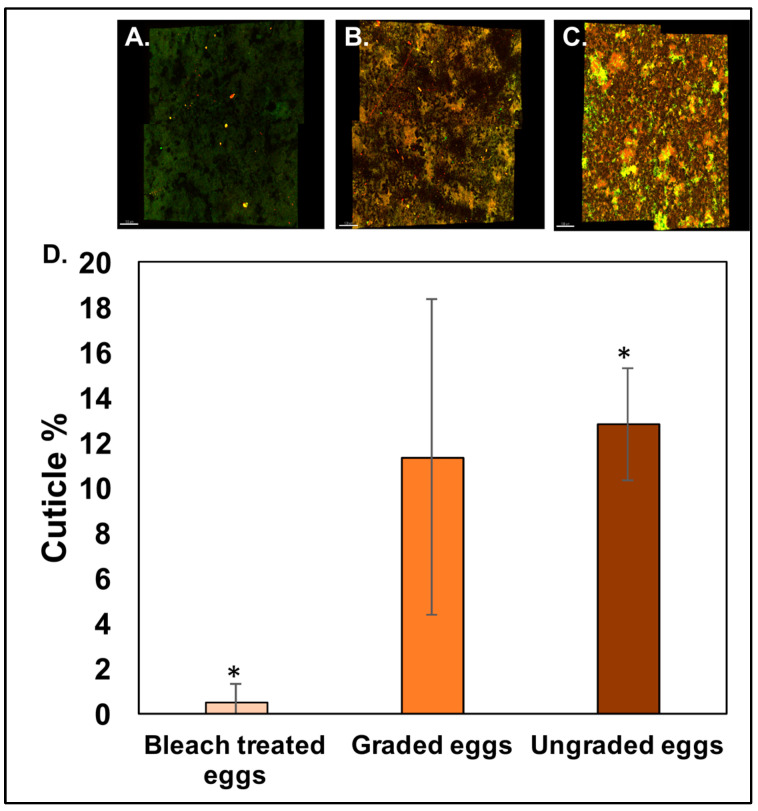
Quantification of cuticle proteins on the eggshell surface. Confocal fluorescent micrographs showing fluorescent cuticle proteins on the outer surface of (**A**) bleach-treated eggs, (**B**) graded eggs, and (**C**) ungraded eggs. (**D**) The impact of egg washing on the percentage (%) cuticle coverage on the outer surface of eggshells displays a lesser coverage in the shells from bleach-treated eggs as compared to the eggshells of ungraded eggs. The significant differences between bleach-treated vs ungraded eggs are represented by (*). Values are represented as mean ± standard deviation (*n* = 3 × 4 = 12, one-way ANOVA; *p* < 0.05).

**Figure 4 microorganisms-12-02027-f004:**
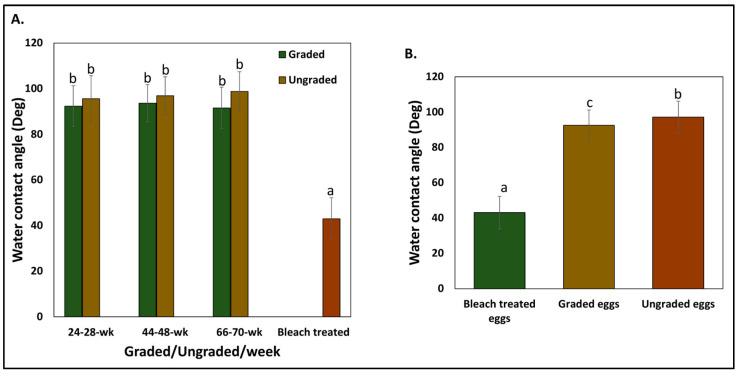
Eggshell surface properties. (**A**) comparison of the contact angle measurements on the surface of eggshells of eggs laid by hens from different ages or bleach-treated eggs (no cuticle) (*n* = 24, ANOVA, Tukey-multiple means comparison; *p* < 0.05) and (**B**) A comparison of the contact angle measurements on the eggshell surface between eggs that were graded/ungraded eggs compared to bleach-treated eggs. The contact angle of the surface of the bleach-treated eggs was lower than graded or ungraded eggs. The values with different superscript letters (Tukey multiple means comparison) are significantly different (ANOVA, Tukey multiple-means comparison; *p* < 0.05). The values represent the mean ± standard deviation (*n* = 72).

**Figure 5 microorganisms-12-02027-f005:**
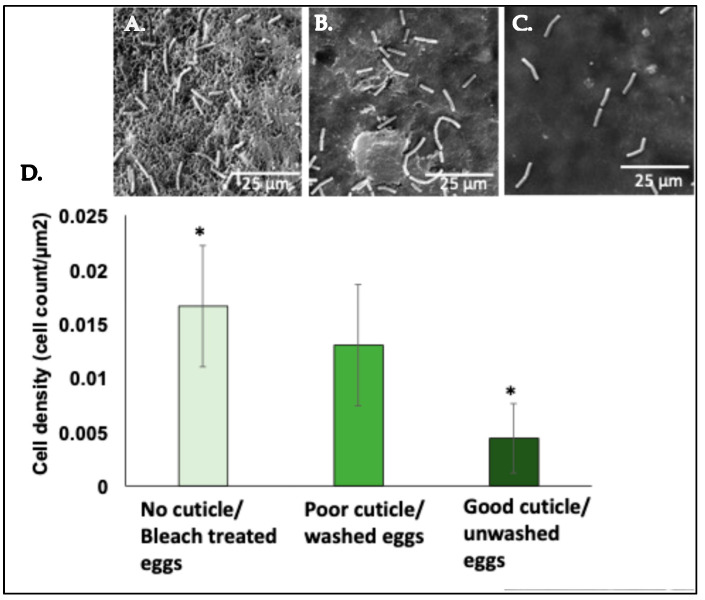
Eggshell surface bacterial adherence. Comparison of *B. cereus* cell density on the outer surface of eggshells. SEM micrographs showing *B. cereus* cells adhered to (**A**) no-cuticle/bleach-treated eggs, (**B**) poor-cuticle/graded eggs, and (**C**) good-cuticle/ungraded eggs. (**D**) Histogram showing abundance (cell count/µm^2^) of *B. cereus* attached to eggshell after 3 h of incubation. The bleach-treated eggshells without cuticle showed a higher abundance of adhering *B. cereus* cells on the outer eggshell surface, in comparison to an intact cuticle on eggshells from ungraded eggs. The significant differences between bleach-treated vs. ungraded eggs are represented by (*).Values represent mean ± standard deviation (*n* = 6 × 3 = 18, one-way ANOVA; *p* < 0.05).

**Figure 6 microorganisms-12-02027-f006:**
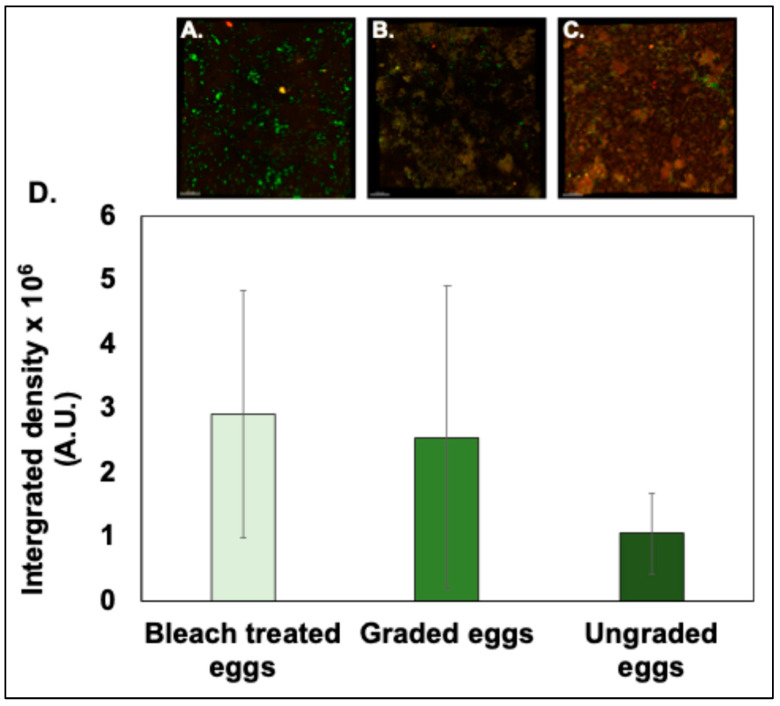
Comparison of *S. typhimurium* density on the outer surface of eggshells. Confocal fluorescent micrographs showing *S. typhimurium* cells adhered to (**A**) bleach-treated eggs, (**B**) graded eggs, and (**C**) ungraded eggs. (**D**) Histogram showing abundance (integrated density A.U.) of *S. typhimurium* attached to eggshells after 3 h of incubation. Values represent mean ± standard error (*n* = 4 × 3 = 12).

**Figure 7 microorganisms-12-02027-f007:**
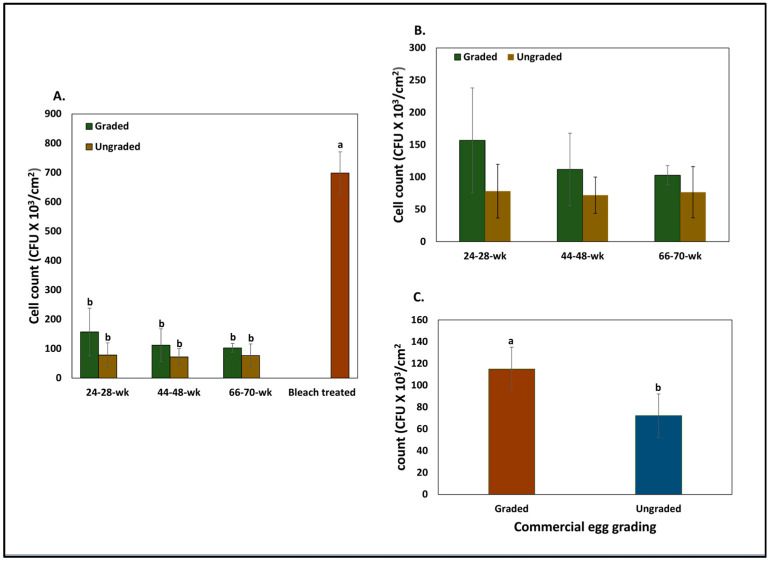
The adherence of *S. typhimurium* to eggshell surfaces. (**A**) The effect of the cuticle removal on *Salmonella* adherence on the eggshell surface. The *Salmonella* cell counts on the outer surface of the no-cuticle (bleach-treated) eggshell were significantly higher than eggshells with an intact cuticle of eggs from hens at different ages. (**B**) The impact of the hen age on *Salmonella’s* adherence to the eggshells’ surface. The values with different superscript letters (Tukey multiple means comparison) are significantly different. The values represent the mean ± SD from six independent samples each performed in triplicate (*n* =18, ANOVA, Tukey multiple-means comparison; *p* < 0.05). (**C**) Effect of the egg washing on *Salmonella’s* adherence to the eggshells’ surface. The values with different superscript letters (Tukey multiple means comparison) are significantly different. The values represent the mean ± standard deviation (*n* = 54, Student’s *t*-test; *p* < 0.05).

**Figure 8 microorganisms-12-02027-f008:**
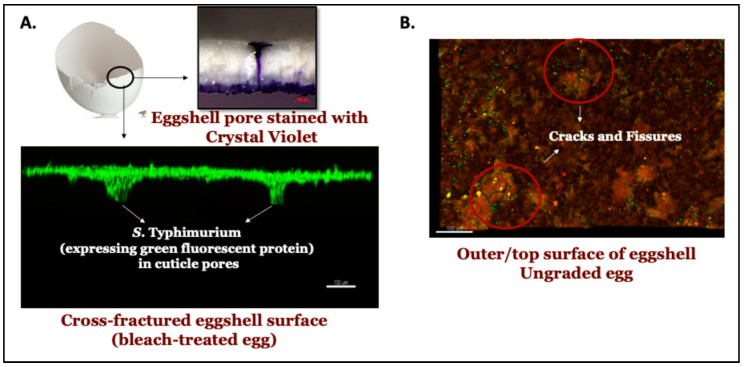
Localization of *S. typhimurium* density on (**A**) a cross-fractured eggshell surface of a bleach-treated egg and (**B**) the outer/top surface of the eggshell of the ungraded egg.

## Data Availability

No new data were created or analyzed in this study. Data sharing is not applicable to this article.

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
