# Peer review of "Effect of Egg Washing and Hen Age on Cuticle Quality and Bacterial Adherence in Table Eggs"

_microorganisms, 2024, doi:10.3390/microorganisms12102027_

Round 1

Reviewer 1 Report

Comments and Suggestions for Authors

The manuscript entitled " Effect of egg washing and hen age on cuticle quality and bacterial adherence in table eggs” by Kulshreshtha et al. is a good quality manuscript and suitable for publication in Microorganisms. In this manuscript, the author conducted systematic investigations and data was presented nicely with good-quality figures. The author also discussed the results adequately. However, author needs to address the following comments before it gets accepted for publication.

Comments to author

1.       The plagiarism report indicates that 19% similarity. The major concern is lines 50-53, 120-124, 433-437 where similarity is detected continuously. The author needs to address it. Also, try to reduce overall plagiarism by 15% or below.

2.       Line 22; correct as 24-28 weeks and 44-48 weeks

3.       The names of bacteria or microorganism should be in italic style

4.       Abbreviations need to be disclosed at their first appearance in the text. Example CDC

5.       The visuals of Figure 1 could be better if the author could use Biorender or similar software.

6.       There is an error in lines 203 and 204. Need to correct it

7.       The author needs to explain why there is a very high standard deviation for Graded eggs in Figure 3.

8.       In Figure 5, the y-axis title seems µm2. Need to address it

9.       The author frequently cited several references together. Examples 308, 309, 323, 362, and so on. It is suggested to site more suitable or find the most suitable in literature and site. A maximum 2 or 3 is recommended.

Author Response

Reviewer 1: The manuscript entitled " Effect of egg washing and hen age on cuticle quality and bacterial adherence in table eggs” by Kulshreshtha et al. is a good quality manuscript and suitable for publication in Microorganisms. In this manuscript, the author conducted systematic investigations and data was presented nicely with good-quality figures. The author also discussed the results adequately. However, author needs to address the following comments before it gets accepted for publication.

1. The plagiarism report indicates that 19% similarity. The major concern is lines 50-53, 120-124, 433-437 where similarity is detected continuously. The author needs to address it. Also, try to reduce overall plagiarism by 15% or below.

Response: We thank the reviewer for the constructive comments. The manuscript has been revised, and we have rephrased and added new content to the text of the manuscript specifically in Lines 50-53, 120-124 to eliminate any similarities with the previous work.

Some similarities, for example, in Lines 433-437 (authors’ contributions and funding) or authors’ address, are unavoidable because our list of authors, their contributions and funding sources are the same as in our previously published two articles on the eggshell cuticle: 1. MPDI: Foods: doi: 10.3390/foods10112559.; 2. Poultry Science: doi.org/10.3382/ps/pex409.

2. Line 22; correct as 24-28 weeks and 44-48 weeks

Response: As suggested by the reviewer, we have included the changes in Line 22.

3. The names of bacteria or microorganism should be in italic style

Response: We thank the reviewer for this valid comment. As suggested, the scientific names of microorganisms have been italicized throughout the manuscript.

4. Abbreviations need to be disclosed at their first appearance in the text. Example CDC

Response: As suggested, the abbreviation has been explained in Line 51.

5. The visuals of Figure 1 could be better if the author could use Biorender or similar software.

Response: We thank the reviewer for this valid comment. As suggested, the image quality is improved using Adobe Illustrator® and an edited version of the image is added to the manuscript.

6. There is an error in lines 203 and 204. Need to correct it

Response: As suggested, the error has been corrected.

7. The author needs to explain why there is a very high standard deviation for Graded eggs in Figure 3.

Response: The percentage of cuticle protein on the outer surface of eggshell from graded eggs showed a high standard deviation, which could be due to variability in the amount of cuticle present or deposited during the production and processing of table eggs. Previous literature, including our study, has shown that cuticle deposition on the outer surface of eggshell is not uniform and can be variable between eggs (Leleu et al., 2011; Rodriguez-Navarro et al., 2013; Bain et al., 2013). Moreover, hen age, commercial egg washing, hen strain, and housing systems can also affect cuticle deposition (Leleu et al., 2011; Samiullah et al., 2017; Kulshreshtha et al., 2018). Hence, the deposition of cuticle proteins on the outer surface could be variable from one egg to another, which could contribute to higher variability in graded eggs. This information has been added to Lines 384-389 for clarification.

References:

- Rodriguez-Navarro, A.; Dominguez-Gasca, N.; Munoz, A.; Ortega-Huertas, M. Change in the Chicken Eggshell Cuticle with Hen Age and Egg Freshness. Poult Sci 2013, 92, 3026–3035.

-Leleu, S.; Messens, W.; de Reu, K.; de Preter, S.; Herman, L.; Heyndrickx, M.; de Baerdemaeker, J.; Michiels, C.W.; Bain, M. Effect of Egg Washing on the Cuticle Quality of Brown and White Table Eggs. Journal of food protection 2011, 74, 1649–54.

-Bain, M.; McDade, K.; Burchmore, R.; Law, A.; Wilson, P.; Schmutz, M.; Preisinger, R.; Dunn, I. Enhancing the Egg’s Natural Defence against Bacterial Penetration by Increasing Cuticle Deposition. Animal genetics 2013, 44, 661–668.

-Samiullah, S.; Omar, A.S.; Roberts, J.; Chousalkar, K. Effect of Production System and Flock Age on Eggshell and Egg Internal Quality Measurements. Poultry Science 2017, 96, 246–258.

- Kulshreshtha, G.; Rodriguez-Navarro, A.; Sanchez-Rodriguez, E.; Diep, T.; Hincke, M.T. Cuticle and Pore Plug Properties in the Table Egg. Poultry Science 2018, 97, 1382–1390.

8. In Figure 5, the y-axis title seems µm2. Need to address it

Response: A cell attachment assay was performed to determine whether cuticle on eggshell surface prevents adherence of Bacillus cereus cells using a previously established method by D’Alba et al., 2013 and 2017. Briefly, small squares (1 cm2) of eggshells were immersed in bacterial suspension in PBS (1 ml, optical density OD = 0.2; ∼108 CFU ml−1 at 600 nm) for 3h at 37 °C. Next, the shell fragments were gently rinsed with PBS to remove excess / non-adhering Bacillus cells. All samples were then fixed with 4% PFA for 10 minutes at 4 °C and the number of adhering bacterial cells was counted (SEM; TeScan Vega-II XMU, Brno—Kohoutovice, Czech Re-public). Bacterial cell counts were performed on five random locations across each shell piece and results were represented as bacterial cell density per μm2 of the shell surface. This information is included in Lines 204-214.

References:

- D’Alba, L.; Torres, R.; Waterhouse, G.I.N.; Eliason, C.; Hauber, M.E.; Shawkey, M.D. What Does the Eggshell Cuticle Do? A Functional Comparison of Avian Eggshell Cuticles. Physiological and Biochemical Zoology 2017, 90, 588–599, doi:10.1086/693434.

-D’Alba, L.; Jones, D.N.; Eliason, C.; Badawy, H.T.; Shawkey, M.D. Antimicrobial Properties of a Nanostructured Eggshell from a Compost-Nesting Bird. Journal of Experimental Biology 2013, 217, 1116–1121, doi:10.1242/jeb.098343.

 9. The author frequently cited several references together. Examples 308, 309, 323, 362, and so on. It is suggested to site more suitable or find the most suitable in literature and site. A maximum 2 or 3 is recommended.

Response: As suggested by the reviewer, the number of references has been reduced throughout the manuscript, especially in Lines 308, 309, 323, 362 and the most suitable 2 or 3 references have been cited.

Reviewer 2 Report

Comments and Suggestions for Authors

The article “Effect of egg washing and hen age on cuticle quality and bacterial adherence in table eggs” is devoted to a relevant issue, – the study of the influence of various factors (cuticle treatment and hen age) on the level of bacterial contamination of eggshell surfaces. The authors have developed a novel enumeration method using green fluorescent protein (GFP) to estimate bacterial adhesion to the eggshell surface and quantify cuticle deposition. Overall, the article is highly relevant, scientifically sound, and advanced in this sector of microbiology. Despite the fact that the results presented in the article are of some importance to scientists and can be relevant to the readers, the article contains some shortcomings and therefore requires a minor revision.

-        At the end of the "Introduction" section, clearly indicate the purpose and objectives of your research.

-        Line 121-123 Eggshells were treated with bleach to remove the cuticle and were used as a control in this study. The cuticle was removed from the outer surface of the eggshells by bleach treatment using a standardized protocol as described previously - The same phrase is repeated.

-        In conclusion, indicate the significance of your results for industrial purposes and scientific research, as well as the goals and prospects for further studies.

Author Response

Reviewer 2: The article “Effect of egg washing and hen age on cuticle quality and bacterial adherence in table eggs” is devoted to a relevant issue, – the study of the influence of various factors (cuticle treatment and hen age) on the level of bacterial contamination of eggshell surfaces. The authors have developed a novel enumeration method using green fluorescent protein (GFP) to estimate bacterial adhesion to the eggshell surface and quantify cuticle deposition. Overall, the article is highly relevant, scientifically sound, and advanced in this sector of microbiology. Despite the fact that the results presented in the article are of some importance to scientists and can be relevant to the readers, the article contains some shortcomings and therefore requires a minor revision.

1. At the end of the "Introduction" section, clearly indicate the purpose and objectives of your research.

Response: We thank the reviewer for the constructive comments. We have stated the purpose and objective of our research at the end of the introduction in Lines 93-98.

2. Line 121-123 Eggshells were treated with bleach to remove the cuticle and were used as a control in this study. The cuticle was removed from the outer surface of the eggshells by bleach treatment using a standardized protocol as described previously - The same phrase is repeated.

Response: We thank the reviewer for this valid comment. The redundancy has been removed.

 3. In conclusion, indicate the significance of your results for industrial purposes and scientific research, as well as the goals and prospects for further studies.

Response: The significance of the work is described in Lines 461-466, and prospects for further studies are summarized in Lines 468-469, in the conclusion.

Reviewer 3 Report

Comments and Suggestions for Authors

This study describes novel strategies to enumerate and localize bacteria associated with eggshell surfaces.   The results can be demonstrated that hen age and commercial egg washing does not significantly impact bacterial adherence. 

Although in the United States and Canada, table eggs are commercially washed before sale;   the findings should be not enough to maintain the confidence of consumers in eggs produced in US and Canada without shelf life time limited.

All the results have not established the safety relationship with shelf life time.

Author Response

Reviewer 3: This study describes novel strategies to enumerate and localize bacteria associated with eggshell surfaces. The results can be demonstrated that hen age and commercial egg washing does not significantly impact bacterial adherence.

Although in the United States and Canada, table eggs are commercially washed before sale;   the findings should be not enough to maintain the confidence of consumers in eggs produced in US and Canada without shelf life time limited.

All the results have not established the safety relationship with shelf life time.

Response. We thank the reviewer for the constructive comments. This work is the third installment of our previously published two research articles on the eggshell cuticle: 1. MPDI: Foods: doi: 10.3390/foods10112559.; 2. Poultry Science: doi.org/10.3382/ps/pex409. In our previous studies, we demonstrated that the industrial washing process can remove some of the cuticle from the outer eggshell surface; nevertheless, the cuticle plug proteins which cover the pore openings remain intact in order to block bacterial entry via the respiratory pores. Our elemental analysis showed that the pore inner surface is rich in phosphorus and chemically distinct from the bulk eggshell mineral. We have also validated that cuticle chemical composition is affected by hen age, strain, housing system and egg-washing (Lines 85-91). Furthermore, in this article, we have shown the effect of egg washing and hen age on bacterial adherence in table eggs. The results validated that hen age (up to 70 weeks) and commercial egg washing do not significantly impact bacterial adherence on eggshell surfaces (Lines 24-25). We did not observe the large effects of increasing hen age or commercial egg washing on bacterial adherence or surface hydrophobicity on the outer surface eggshell. However, adhering bacterial cells on the outer cuticle surface were significantly lower in all conditions when compared to bleached treated eggs (with no cuticle). Also, in our previous study, we observed that cuticle quality declined with increasing hen age or due to commercial washing.  Hence, our findings collectively indicate that some cuticle components can be thinner due to increasing hen age or wear away during the commercial washing process. However, the constituent antimicrobial proteins could still be sufficiently abundant and functional / active to block pathogens from adhering to the outer surface of the eggshell irrespective of hen age or egg washing, thus protecting the egg from bacterial contamination. The antimicrobial cuticle proteins are possibly more stable to the above traits and can compensate for cuticle loss in order to provide better protection to the egg against invading pathogens.

Round 2

Reviewer 1 Report

Comments and Suggestions for Authors

As the author addressed the comments. the current version of manuscript recommended to accept for publication